# HIF-1α and HIF-2α Differently Regulate the Radiation Sensitivity of NSCLC Cells

**DOI:** 10.3390/cells8010045

**Published:** 2019-01-12

**Authors:** Eloy Moreno Roig, Arjan J. Groot, Ala Yaromina, Tessa C. Hendrickx, Lydie M. O. Barbeau, Lorena Giuranno, Glenn Dams, Jonathan Ient, Veronica Olivo Pimentel, Marike W. van Gisbergen, Ludwig J. Dubois, Marc A. Vooijs

**Affiliations:** Department of Radiotherapy (MAASTRO), GROW—School for Oncology and Developmental Biology, Maastricht University, 6229 ET Maastricht, The Netherlands; e.morenoroig@maastrichtuniversity.nl (E.M.R.); arjan.groot@maastrichtuniversity.nl (A.J.G.); ala.yaromina@maastrichtuniversity.nl (A.Y.); t.hendrickx@maastrichtuniversity.nl (T.C.H.); l.barbeau@maastrichtuniversity.nl (L.M.O.B.); l.giuranno@maastrichtuniversity.nl (L.G.); g.dams@maastrichtuniversity.nl (G.D.); j.ient@maastrichtuniversity.nl (J.I.); v.olivopimentel@maastrichtuniversity.nl (V.O.P.); m.vangisbergen@maastrichtuniversity.nl (M.W.v.G.); ludwig.dubois@maastrichtuniversity.nl (L.J.D.)

**Keywords:** hypoxia, HIF, radiotherapy, metabolism

## Abstract

The hypoxia-inducible transcription factors (HIF)-1/2α are the main oxygen sensors which regulate the adaptation to intratumoral hypoxia. The aim of this study was to assess the role of the HIF proteins in regulating the radiation response of a non-small cell lung cancer (NSCLC) in vitro model. To directly assess the unique and overlapping functions of HIF-1α and HIF-2α, we use CRISPR gene-editing to generate isogenic H1299 non-small cell lung carcinoma cells lacking HIF-1α, HIF-2α or both. We found that in HIF1 knockout cells, HIF-2α was strongly induced by hypoxia compared to wild type but the reverse was not seen in HIF2 knockout cells. Cells lacking HIF-1α were more radiation resistant than HIF2 knockout and wildtype cells upon hypoxia, which was associated with a reduced recruitment of γH2AX foci directly after irradiation and not due to differences in proliferation. Conversely, double-HIF1/2 knockout cells were most radiation sensitive and had increased γH2AX recruitment and cell cycle delay. Compensatory HIF-2α activity in HIF1 knockout cells is the main cause of this radioprotective effect. Under hypoxia, HIF1 knockout cells uniquely had a strong increase in lactate production and decrease in extracellular pH. Using genetically identical HIF-α isoform-deficient cells we identified a strong radiosensitizing of HIF1, but not of HIF2, which was associated with a reduced extracellular pH and reduced glycolysis.

## 1. Introduction

Regions of oxygen deprivation (hypoxia) are a common feature of many solid cancers and are caused by an aberrant tumor vasculature. Hypoxic tumor cells activate stress response pathways in order to adapt to these low oxygen levels [1]. Within the tumor, some cells react to hypoxic stress with adaptive responses that, by changing their gene expression, confer an aggressive phenotype and resistance to therapy [2]. Importantly, hypoxic tumors are more resistant to chemotherapy because they are poorly perfused, proliferate slower and also upregulate drug efflux pumps. Hypoxic tumor cells are also more radiation resistant. Decreased treatment sensitivity of hypoxic tumor cells leads to chemo-radiotherapy treatment failure in cancer patients [3,4]. Despite the limited success of hypoxia modification therapy strategies in several phase 3 clinical trials, hypoxia is still a promising tumor-selective therapeutic target.

Mammalian cells encode for three hypoxia inducible factors (HIF)-α orthologs (HIF1–3), which have all been demonstrated to be oxygen-regulated. HIFs are bHLH-PAS domain heterodimers composed of a constitutively expressed β-subunit and an O_2_-regulated α-subunit, which is hydroxylated at two proline residues under aerobic conditions by O_2_-dependent prolyl hydroxylase domain proteins (PHDs) [5]. When oxygen is present, prolyl-hydroxlylated HIF-α undergoes rapid ubiquitination and subsequent proteasomal degradation involving the von Hippel−Lindau protein (pVHL), which is a member of the E3 ubiquitin protein ligase family. Under hypoxic conditions, the O_2_-dependent PHD/FIH are inhibited, HIF-α accumulate, forms a heterodimer with HIF-β and induces transcriptional activation of canonical HIF target genes such as carbonic anhydrase IX (CAIX), glucose transporter 1 (GLUT1) and vascular endothelial growth factor (VEGF) [6]. HIF-1α and HIF-2α proteins bind to the same common hypoxia response elements (HRE) [7], but their transcriptional activities are determined by multiple factors including the presence of repressors, differential sensitivity to HIF hydroxylation and differential recruitment of co-activators [8]. In particular, differences in tissue-specific and temporal induction patterns of each isoform suggest that HIF-1α and HIF-2α may have distinct transcriptional targets that regulate specific aspects of hypoxic signaling and adaptation in different cell types [9]. This specific gene expression profile in hypoxic cells by either HIF-1α and/or HIF-2α can contribute differently to a malignant phenotype in cancer cells [10]. HIF stabilization is frequently observed in cancers associated with a malignant phenotype and is an important prognostic factor for poor outcomes in most solid cancers [11].

Preclinical studies have demonstrated the potential to pharmacologically block both HIF-1α and/or HIF-2α, causing tumor growth delay and improved local control [12]. Pharmacological inhibition of HIF-1α increases the therapeutic efficacy of radiation treatment both in vitro and in vivo [13,14]. In particular, cells lacking HIF-1α are markedly more sensitive to altered redox activity and glucose deprivation, suggesting a protective role upon stress conditions in tumors [15]. The metabolic switch towards increased glycolysis in hypoxic environments also affects the ability of these cells to survive therapy. For instance, the enhanced glycolytic rate increases the intracellular amounts of lactate and CO_2_, which are transported out of the cell, contributing to an extracellular acidic pH [16] and an increased therapy resistance [17]. The substantial increase of lactate production during hypoxia allows the scavenging of reactive oxygen species (ROS) molecules generated after radiation exposure due to the antioxidant properties of lactate [18]. HIF proteins are important regulators of the tumorigenic capacity and therapy sensitivity of cancer cells; however, the regulatory mechanism of HIF-1/2α mediating the response of non-small cell like cancer (NSCLC) cells to radiation therapy needs further investigation. Previous studies have shown that the expression of HIF-1α and HIF-2α differently impact on patient prognosis and other clinicopathological characteristics in several cancers, such as human NSCLC [19]. Moreover, HIF-2α is highly expressed in cancer stem cells, which has been associated with a radioresistant phenotype in lung cancer. Consequently, HIF-2α negatively correlates with prognosis in patients treated with radiation [20].

Taken together, mounting evidence implicates both HIF-1α and HIF-2α proteins in a highly context dependent but orchestrated manner in tumor cell survival, hypoxia tolerance and treatment response. Here we used CRISPR/Cas9 gene-editing to generate loss-of-function deletions in HIF-1α, HIF-2α, or both HIF-1α and HIF-2α in H1299 NSCLC cells. Using these cells, we assessed the contribution of HIF-1α and HIF-2α factors as important mediators of hypoxia response, radiation response and metabolic adaptation.

## 2. Materials and Methods

### 2.1. Cell Culture and Hypoxia Exposure

Human non-small cell lung carcinoma (NSCLC) H1299 (ATCC CRL-5803) cells were grown in RPMI 1640 (Westburg, Leusden, The Netherlands) supplemented with 10% FBS (Sigma-Aldrich, St Louis, MO, USA), penicillin (10 U/mL) and streptomycin (10 μg/mL). Normoxic cells were incubated in a humidified atmosphere with 5% CO2/21% O_2_ at 37 °C. Cells were exposed to hypoxic conditions in a hypoxic chamber (MACS VA500 microaerophilic workstation, Don Whitley Scientific, Bingley, UK) to stabilize HIF proteins. The composition of the atmosphere in the incubator consisted of 5% CO_2_, 0.2% O_2_ and residual N_2_. The identity of H1299 cells was verified by STR genotyping (Identicell, Aarhus, Denmark).

### 2.2. CRISPR/Cas9-Mediated Knockout of Human HIF1α and HIF2α Genes

In order to generate HIF-1α and HIF-2α knockouts, H1299 cells were transfected with CRISPR/Cas9 using pSpCas9(BB)-2A-GFP (PX458) or pSpCas9(BB)-2A-Puro (PX459), a gift from Feng Zhang (Addgene plasmid #48139, Watertown, MA, USA) [21]. Two small guide RNA (sgRNA) targeting the HIF-1α and HIF-2α locus were designed using the CRISPR Design Tool (http://crispr.mit.edu). Transfections were performed with Lipofectamine 2000 (Thermo Fisher, Waltham, MA, USA) and Puromycin resistant cells were detected after 3 days of puromycin selection (1 μg/mL) followed by a recovery period of two weeks. Surviving cells were single cell seeded and expanded for further analysis. Genomic DNA was extracted from the potential candidates and the targeted regions of the HIF-1α and HIF-2α genes were PCR amplified. The product was cloned into TOPO-TA vector to isolate every single allele from the whole cell population. DNA fragments were sequenced and analyzed for further selection of total HIF-1α and HIF-2α knockout clones. Multiple independent H1299 cell lines carrying mutations in HIF-1α (H1KO) and HIF-2α (H2KO) genes were established. Additionally, the H1KO-B5 clone was used as a cellular template to target the HIF-2α proteins and thus create a double knockout cell line (dHKO) (Appendix A).

### 2.3. Western Blotting and Real Time Quantitative PCR

SDS–PAGE, Western blot and qRT-PCR were performed according to standard protocols. Total RNA from cells was isolated using RNeasy (Qiagen, Venlo, The Netherlands) and Nucleospin RNAII (BIOKE, Leiden, The Netherlands), respectively. Antibodies used are listed in Appendix A. Primers used are listed in Appendix A.

### 2.4. Proliferation and Clonogenic Survival Assay

Proliferation was monitored during 2 days by counting cells in an automatic cell counter. For clonogenic survival, cells were counted and seeded on day 0. Cells were placed under 0.2% O_2_ the day after for 24 h. Irradiation was performed using a 225 kV Philips X-ray tube on day 2. Subsequently, cells were trypsinized and plated in triplicate for clonogenic survival. Cells were allowed to form colonies for 12 days, fixed and stained with a 0.4% methylene blue (Sigma-Aldrich, St Louis, MO, USA) in 70% ethanol solution. Colonies were defined as >50 cells. Data were fitted using the LQ model.

### 2.5. Cell Viability—Hypoxia Tolerance

Crystal violet staining was performed to assess cell viability. Cells were incubated in 6-well plates at 1.0 × 10^4^ cells per well and cell viability was measured at 5 days after hypoxic incubation. Culture medium was removed, cells were fixed with 4% paraformaldehyde at room temperature for 10 min and then stained with 0.05% crystal violet for 30 min. The cells were then washed with tap water, after which the water was removed and the cells were dried out on filter paper. Blue dye was dissolved in 500 μL of methanol and emission spectra were measured at an excitation wavelength of 540 nm using a Multimode Microplate Reader.

### 2.6. γ-H2AX Immunocytochemistry

Cells were fixed with ice-cold 100% methanol. Subsequently, cells were permeabilized with 0.2% Triton-X solution (in PBS) and normal goat serum was used as blocking agent. Cells were stained with a primary anti-phospho (Ser139)-H2AX antibody (1:500, Millipore, Burlington, MA, USA) followed by anti-rabbit Alexa Fluor 488 (1:500, Invitrogen, Carlsbad, CA, USA) as secondary antibody. Hoechst 33342 (20 μg/mL, Sigma-Aldrich, St Louis, MO, USA) was used as nuclear counter stain.

### 2.7. Image Analysis

For evaluation of γ-H2AX staining, fluorescent images were captured at 64× objective using a Leica TCS SPR confocal microscope. Images were taken in 2 μm planes along the Z-axis and foci analysis was performed in maximal intensity projection Z-stacks using ImageJ software (v. 1.50, National Institutes of Health) in a semi-automated way. Nine images have been acquired per treatment condition resulting in 100 ± 20 cells analyzed in total. First, DAPI positive nuclei were selected using a manual segmentation procedure based on signal intensity and background staining and nucleus area was recorded. Only complete intact nuclei per image field were evaluated. Second, raw integrated density (RID) of γ-H2AX per nucleus, that is, the sum of the gray values of the pixels, was calculated. To correct for variability in nucleus size, RID was normalized to the nucleus area for each cell. Data are presented as mean ± SD of normalized RID per cell line and treatment group. To validate the method for γ- H2AX quantification, γ-H2AX foci were manually counted in randomly selected images in a manner blinded to the cell line and treatment group. Comparison of γ-H2AX foci with normalized RID shows significant correlation (r = 0.9608, *p* < 0.001) indicating that normalized RID reflects the number of γ-H2AX foci.

### 2.8. Cell Cycle Analysis

For cell cycle analysis, cells were incubated either under normoxic or hypoxic conditions for 24 h, exposed to radiation and placed under normoxia for 4 h. Cells were washed with PBS, treated with trypsin and fixed in ice-cold 70% ethanol for at least 24 h. Before analysis, cells were washed with PBS and stained with propidium iodide (PI) for 30 min at room temperature. Analysis was performed using a FACS CANTO II. Data obtained from the cell cycle distributions were analyzed using a FlowJo_10.

### 2.9. pH and Extracellular L-Lactic Acid Measurements

Changes in extracellular pH were monitored using a pH meter (Beckman Coulter, Brea, CA, USA, pH 350). Cells were seeded at different cell numbers and incubated for 24 h under 0.2% O_2_. Levels of extracellular L-Lactic acid were measured using the L-Lactic acid kit (Biosentec, Toulouse, France) according to manufacturer’s guidelines. Both pH and L-Lactic acid levels were corrected for cell counts.

### 2.10. Metabolic Profiling

Cells were seeded at an optimized cell density of 3 × 10^4^ cells/well. Metabolic profiles were generated by replacing the growth medium for assay media 1 h before using the Seahorse XF96 extracellular Flux analyzer (Seahorse Bioscience, Billerica, MA, USA) according to manufacturer’s guidelines.

### 2.11. Statistics

All assays were performed at least three times, and results are expressed as means ± standard deviations. Analyses were performed with GraphPad Prism 5. Statistical tests were always performed relative to WT cells. Unpaired two-tailed Student’s *t*-tests or Mann-Whitney U tests were performed to determine significance. A sum-square test for curve comparison was performed to determine significance of clonogenic data. *p* values <0.05 were considered significant.

## 3. Results

To examine the radiobiological and metabolic properties of HIF-1α and HIF-2α, we generated HIF loss-of-function mutants in H1299 cells using the type II CRISPR/Cas9 system. Single allele sequencing confirmed that cells carried mutations that led to premature termination of the HIF-α open reading frame. Each knockout harbored two or three different mutated alleles leading to one or several STOP codons (Appendix A). We verified that H1299 clones did not have the Cas9 plasmid integrated (data not shown). Western blotting confirmed the absence of HIF proteins (Figure 1A). We observed a prominent increase in HIF-2α stabilization following hypoxia incubation in H1KO cells, but without elevated HIF-2α mRNA expression levels (Appendix A). On the contrary, HIF-2α-deficiency did not influence the hypoxic induction of HIF-1α protein expression. The overall expression levels of HIF-1β were decreased in all the knockout models in comparison with WT cells (Figure 1A). Next, we determined the mRNA expression levels of the canonical hypoxia-induced genes CAIX, GLUT1, CITED2 and TWIST1. We observed that the induction of these genes was severely compromised in the absence of HIF-1α and/or HIF-2α proteins under hypoxia (Figure 1B). Furthermore, only small differences were seen in the proliferative capacity of single HIF mutants in comparison with WT cells, both under normoxic and low oxygen conditions. In dHKO cells, a small but significant (*p* = 0.0124) growth delay was observed compared to wildtype cells under normoxic conditions (Figure 1C) and under prolonged hypoxic conditions (*p* = 0.0494) (Figure 1D).

Next, we assessed the impact of HIF-1/2α deletion on clonogenic survival following irradiation. Under normoxic conditions, no differences in survival were observed between WT and H1KO or H2KO cells, while overall survival was significantly (*p* = 0.0006) reduced for dHKO cells (Figure 2A). Additionally, clonogenic survival assays revealed that under hypoxic conditions the overall surviving fraction of H1KO cells was significantly (*p* < 0.0001) higher than WT cells (Figure 2B). Similar results were observed for two independently derived H1KO and H2KO CRISPR/Cas9 clones (Appendix A). We also observed a more radiosensitive phenotype in dHKO cells irradiated under hypoxic conditions although to a lesser extent.

Next, we investigated whether differences in the DNA damage response (DDR) or DNA repair were responsible for these differences in radiation sensitivity. No DNA damage was observed in the absence of irradiation under normoxic or hypoxic conditions using γ-H2AX staining. Four hours after irradiation under normoxia, γH2AX recruitment increased significantly only in dHKO cells (*p* = 0.0007) (Figure 3A). Upon hypoxia we observed a strong and significant decrease (85%, (*p* = 0.002)) of γH2AX labelling in irradiated H1KO cells compared to wild type and H2KO cells (Figure 3B). γH2AX recruitment was highest in dHKO cells. These differences disappeared 24 h after irradiation, independent of the oxygen concentration. The γH2AX labelling in irradiated cells under hypoxia and normoxia is consistent with the long-term survival outcome of dHKO and H1KO cells.

To identify the immediate effects of HIF-1/2α on the expression and activation of important DDR proteins, we examined the expression of pATM and its downstream target, p(Thr68)-CHK2, 4 h after radiation (Figure 4). Levels of pATM were equally induced upon radiation, independent of the oxygenation status and HIF genotype. Exposure to radiation enhanced pCHK2 levels both upon normoxia and hypoxia for all cells, except for dHKO cells where under normoxia a slight decrease was observed. Strikingly, levels of total and phosphorylated CHK2 protein were upregulated in single and double-HIFKO cells in comparison with WT cells. Similar results were observed in an independent series of experiments (Appendix A).

Flow cytometry was used to determine the effect on cell cycle distribution in irradiated cells lacking HIF-1/2α (Figure 5). dHKO cells showed a slight delay in cell proliferation (Figure 1C), which was supported by a significant reduction in the G1 population. As expected, irradiation induced a G2/M cell cycle arrest in all cells regardless of the presence of HIF. In line with the survival data, this effect was more prominent (*p* = 0.0074) in dHKO in comparison with WT cells. The increase in percentage of cells in G2/M was accompanied by a significant (*p* = 0.0180) decrease in cells in S phase. In addition, WT and HIF-deficient cells, incubated under hypoxic conditions for 24 h, showed an overall decrease in S phase and accumulation in the G2/M phase. Under hypoxia, a decrease (*p* = 0.041) in G1 phase was observed for the dHKO cells in comparison with WT cells, which is consistent with an enhanced (*p* = 0.0297) accumulation in G2/M phase. However, when irradiation was performed under hypoxia, no additional effects on cell cycle distribution were observed.

To determine whether changes in extracellular acidification are involved in the radiation response of HIF-deficient cells under hypoxia, we performed in vitro extracellular pH measurements and assessed lactic acid release into the medium both under normoxic and hypoxic conditions. No differences in extracellular pH and lactic acid concentration were observed under normoxic conditions (Figure 6A, C). After hypoxic incubation the pH dropped for all cellular models, but extracellular pH was significantly (*p* = 0.0017) more reduced in H1KO cells (Figure 6B), with a concomitant increase (*p* = 0.0008) of lactic acid in medium (Figure 6D). In order to investigate whether lactate transporters are differently regulated in the genetic models, we measured the mRNA expression levels of the monocarboxylate transporter 1 (MCT-1) and 4 (MCT-4). While the expression levels of MCT-4 were regulated by hypoxia, MCT-1 levels were unaffected (Figure 6E). The hypoxia-induced expression of MCT4 expression was perturbed in all HIF-deficient cells and reached significance in H1KO cells (Figure 6F).

To investigate the impact of HIF-1/2α deletion on cellular respiratory metabolism, we measured the extracellular acidification rate (ECAR). Basal ECAR was measured under control conditions and in the presence of desferrioxamine (DFO), an iron chelator that stabilizes HIF. Our results demonstrated that HIF-deficient cells did not show differences in ECAR compared to WT cells under control conditions (Appendix A). On the other hand, when cells were treated with DFO, dHKO cells showed a significant (*p* = 0.0286) reduction in basal ECAR compared to WT cells (Appendix A). No differences in mitochondrial respiration were found when assessing basal oxygen consumption rate (OCR) levels under normoxia and hypoxia (Appendix A). Further analyses of the glycolytic profile of the genetic models were performed by measuring ECAR during a glycolytic stress test. As expected, the absence of HIF proteins did not result in changes in ECAR regulation under normoxic conditions (Appendix A). In contrast, H1KO and H2KO cells induced a gain in glycolytic reserve accompanied by reduced glycolytic activity when stimulated with DFO (Appendix A).

## 4. Discussion

Hypoxic cells upregulate the HIF pathway which is associated with radiation resistance in many types of cancer, including NSCLC [22]. Clinically, high levels of HIF-1α and HIF-2α positively correlate with tumor progression and poor patient outcome [19,23]. In the present study, we investigated the role of HIF-1/2α on radiation sensitivity of NSCLC cell line H1299. Therefore, we generated a novel in vitro genetically engineered model of NSCLC with intact or deleted HIF-1/2α proteins. The guide RNAs used to target both HIF-1α and HIF-2α targeted the first and second exon of each gene, respectively. Deletion of HIF-1α caused a compensatory increase in the levels of HIF-2α when oxygen was limited. In contrast, the same effect was not observed in HIF-2α deficient cells, suggesting that HIF-1α is sufficient to trigger the signaling pathways that regulate response to hypoxia. It was previously reported that cancer cells expressing both HIF-1α and HIF-2α proteins might mutually compensate by upregulating HIF-1α under HIF-2α knockdown conditions [24], favoring a malignant phenotype [25]. Conversely, other studies observed that HIF-1α knockdown cells increased HIF-2α protein levels compared with the non-silencing control [26]. The compensatory effect by increased levels of HIF-2α could therefore be cell type specific. In line with our data, the compensatory regulation of HIF proteins can be explained by an enhanced translation activity rather than increased mRNA production [27].

Previous studies have focused on unravelling the differential roles of HIF-1α and HIF-2α in hypoxic gene regulation. Both proteins seem to display distinct, non-overlapping biological functions and independently regulate specific target genes in a cell type specific manner [28]. For instance, hypoxic induction of HIF-1α target genes is repressed in HIF-1α-deficient endothelial cells, suggesting that its loss cannot be compensated by HIF-2α or other hypoxia-inducible proteins [29]. Here we show that the absence of either HIF-1α or HIF-2α is sufficient to block canonical HIF target gene expression, suggesting that both proteins are essential for the activation of the HIF pathway in H1299 NSCLC cells. HIF-1β levels are also reduced in the knockout cells in comparison with HIF-expressing cells. HIF-1β is necessary for the assembly and functionality of an active HIF complex [6], meaning that its downregulation could contribute to the reduction of hypoxia-related gene activation.

Our results raise some interesting questions about how HIF proteins regulate a radiation response in NSCLC cells. First, it was shown that hypoxic HIF-1α-deficient cells exhibit a more radioresistant phenotype than WT cells. This was a surprising finding, especially in light of data from other studies showing that HIF-1α controls hundreds of genes involved in radioprotection in tumors, such as GLUT1 or TWIST [30]. It is widely accepted that HIF-1α contributes to radiation resistance of a large variety of tumor types by inducing cell cycle arrest [31], inhibiting apoptosis [32], promoting angiogenesis [33], enhancing glycolytic metabolism [34] and suppressing ROS production by mitochondria [35]. Stegen et al. have shown that HIF-1α strongly induces the conversion of glutamine into glutathione and activity of other ROS scavenging enzymes to maintain redox balance [36]. Thus, in HIF-1α-deficient H1299 cells, the compensatory activity of HIF2α might be responsible for suppressing ROS and contributing to the increased radiation resistance. In contrast to our finding, other studies have shown that HIF-1α is required for p53 mediated induction after radiation and hypoxia, and is responsible for reduced in clonogenicity [37]. Significantly, H1299 cells are p53-deficient and the effect of HIF2α was not reported in these studies. The increased sensitivity of double knockout HIF-1/2 cells strongly argues that the increase of HIF-2α in HIF-1α deficient cells is responsible for this radiation resistance, although these effects are cell type dependent. To note, only a few studies attempted to understand the biological mechanisms behind HIF-2α regulating radiation response in cancer. A negative prognostic role for HIF-2α expression has been seen in patients with different types of cancer [19] and is also associated with a poor response to radiotherapy [38,39]. Moreover, HIF-2α inhibition enhanced radiation sensitivity in a cellular model of lung cancer by promoting apoptotic activity via the p53 pathway [40]. We therefore hypothesize that the substantial accumulation of HIF-2α seen in hypoxic HIF-1α-deficient cells may be responsible for the radioresistant phenotype. Conversely, there were no differences in clonogenic survival between wild type and HIF-2α-deficient cells treated with radiation, suggesting that HIF-1α is able to compensate the lack of HIF-2α without altering their radiobiological response. Using HIF-1α/2α-deficient cells we demonstrated that the radioprotective effect seen in HIF-1α-deficient cells is dependent on HIF-2α compensation. As expected, deleting both HIF proteins rescued the radioprotective phenotype seen in HIF-1α-deficient cells. Moreover, cells lacking both HIF-1 and HIF-2α displayed a more radiosensitive phenotype in comparison with wildtype cells, indicating an important role of both HIF proteins in the radiation response of tumor cells under normal oxygen conditions and hypoxia. A possible explanation for the increased radiation sensitivity of double knockout HIF cells is the reduced glycolytic profile, which was reflected by the reduction in ECAR upon DFO treatment of double knockout HIF cells compared to wild type. Of note, this did not lead to changes in pH or lactic acid. Due to technical limitations we were not able to measure oxygen consumption or glycolytic activity under hypoxia. It is well established that oncogene activation in cancer cells can stabilize HIF under oxygenated conditions. The best-known examples are the loss of the VHL tumor suppressor gene [41] oncogene activation, such as EGFR [42] and PI3/AKT [43]. In addition, increased levels of succinate and fumarate, competitive inhibitors of α-ketoglutarate, are required for prolyl hydroxylase activity, which may stabilize HIF under normoxic conditions [44,45]. HIF is also activated by reactive oxygen/nitrogen species. Radiation induced revascularization and reoxygenation leads to reactive oxygen species and HIF activation. Under these circumstances HIF is highly radioprotective [37,46]. In contrast, HIF-inhibition post-irradiation in tumors promotes radiation sensitivity [37], while HIF inhibition by itself does not affect tumor growth [37,47].

Based on our findings, we suggest that dual HIF-1α/HIF-2α inhibitors are potentially clinically relevant enhancers of tumor radiosensitivity, but that HIF-1α inhibitors alone may increase radiation resistance. There are many compounds that inhibit HIF function, some of which have been successful in preclinical studies by reducing the hypoxic fraction and increasing radiation sensitivity [48]. However, most of these compounds are a-specific and have been largely unsuccessful in clinical trials [49]. More recently, specific HIF-2α inhibitors have been produced that have shown some clinical efficacy in renal cell carcinoma [50].

It is shown here that in HIF-deficient cells there is a substantial increase in basal and hypoxia induced CHK2 protein levels and phosphorylation without changes in activation of its upstream regulator ATM. Previously, studies described the role of the DDR in relation with specific cellular stress modulators such as hypoxia and HIF. It appears that ATM is active in hypoxia and increases HIF-1α stability through phosphorylation at S^696^, which in turn leads to a reduction in mammalian target of rapamycin complex 1 (mTORC1) signaling [51]. Given the high degree of crosstalk between ATM and hypoxia, and the role of HIF in cell cycle regulation, it is perhaps not surprising that CHK2 can be modulated by HIF expression. Our finding that CHK2 is strongly activated in cells lacking both HIF supports the strong G2/M and G1 arrest in these cells upon hypoxia. Further research is needed to better understand the link between these proteins. Interestingly, the activation of the DDR in relation with the status of HIF did not correlate with our cell survival outcome and double-strand break (DSB) formation. ATM is considered the major regulator of H2AX protein phosphorylation in response to DSB formation, which consequently leads to cell cycle arrest and cell death [52]. Our data demonstrates that ATM regulation is not responsible for the differences seen in radiation sensitivity of the HIF-deficient cells versus WT cells, suggesting an alternative mode of HIF regulating radiation response in cells. Importantly, it has been demonstrated that p53-deficient cells, such as H1299, rely on a different cell-cycle checkpoint pathway involving p38MAPK/MK2 for cell-cycle arrest and survival after DNA damage [53]. Furthermore, the influence of combined HIF-1α and HIF-2α subunits on radiation response was further validated by analyzing the cell cycle distribution. The findings reported here indicate that irradiated dHKO cells undergo a higher G2/M phase arrest than control cells. Notably, this effect was only significant when irradiation was performed under normoxic conditions. Other studies have also observed an increased cell cycle arrest in irradiated cells under hypoxia when deleting HIF-1α [31] and HIF-2α [40], whereas our data show that the inhibition of both proteins is necessary for an increased cell cycle abrogation. Moreover, in line with the hypoxia tolerance experiments, in cells lacking both HIF-1α and HIF-2α a strong decrease in S-phase and significant increase in G2/M arrest was observed. These data point towards both complementary and overlapping activities of HIF in hypoxic response and are likely context dependent.

To date, multiple lines of evidence support that certain metabolites, such as lactate, confer cells with a malignant behavior and a pro-metastatic phenotype in cancer patients, resulting in a potential prognostic biomarker in cancer [54]. Tarnawski et al. have shown that lactate levels detected in glioma tumors are a prognostic parameter for predicting recurrence and overall survival after postoperative radiotherapy [55]. Lactate regulates different biological activities within the cell, which ultimately lead to an increased angiogenesis capacity, antioxidative and immunosuppressive effects, contributing to tumor evasion and resistance to therapy [56]. Moreover, decreasing the levels of lactate in tumors enhances response to high-dose single-fraction radiotherapy in solid tumors [57]. Previous investigations have demonstrated the free radical scavenging activity of lactate; however, the mechanisms underlying such antioxidant effect have yet to be elucidated [18]. The high levels of lactic acid and consequently low pH seen in the medium of HIF-1α deficient cells are a potential explanation for the radioresistant phenotype seen in these cells. The specific mechanisms explaining the increased lactate secretion in these cells need further investigation. We hypothesize that the compensatory upregulation of HIF-2α proteins in HIF-1α deficient cells promotes glycolysis and an increased lactate production and radioresistant phenotype. Future experiments should examine the individual contribution of HIF-2α in mediating these effects and may be a viable target for therapeutic interventions. Despite this, the lactate transporter MCT-4 was strongly downregulated in HIF-1α deficient cells, suggesting that elevated extracellular lactic acid levels are not caused by increased expression of MCT-4. Nonetheless, other studies previously reported anti-proliferative effects when blocking lactate influx by MCT-1 and MCT-4, blocking the capacity to take up lactate as an energetic fuel [58]. Therefore, reduced levels of the lactate transporter MCT-4 might also influence the capacity of H1299 cells to recycle extracellular lactate back into cells and thus cause its accumulation. The same phenotype was seen when investigating other hypoxia-related genes involving pH and metabolite balance such as CAIX and GLUT1. Although the expression of MCT4 was downregulated, it is possible that its activity was enhanced, hence the increase in lactate; however, this explanation remains speculative. Here, we also show that combined HIF-1/2α deficiency caused a substantial decrease in the baseline ECAR, as a result of non-glycolytic acidification, possibly contributing to the enhanced radiation sensitivity observed in dHKO cells. However, the real glycolytic activity measured in DFO-treated cells was diminished by HIF-1α deficiency in comparison with the other genetic models. Previous studies demonstrated that cells with increased lactic acidosis and higher proton density became less glycolytic by impairing several enzymes of the glycolytic pathway [59]. The above results suggest that HIF-1α-deficient cells have a defect in the glycolytic pathway, possibly caused by elevated lactate levels, which in turn leads to a more radioresistant phenotype. However, the specific mechanism regarding the role of HIF-2α in regulating the levels of lactate needs further investigation.

Overall, our work may have relevant implications concerning how HIF-1α inhibitors are currently used in the clinic. Our data suggest that HIF-1α inhibition in combination with radiation might increase radiation resistance. As an alternative, we propose that HIF-1α blockade should be used concurrently with the inhibition of HIF-2α in order to radiosensitize tumor cells, partly by reducing their basal-glycolytic activity. Importantly, the specific inhibition of HIF-1α accounts for critical changes in the metabolic profile of H1299 cells, including an increase in extracellular lactate which might result in a defect in using glucose as an energetic fuel.

## Figures and Tables

**Figure 1 cells-08-00045-f001:**
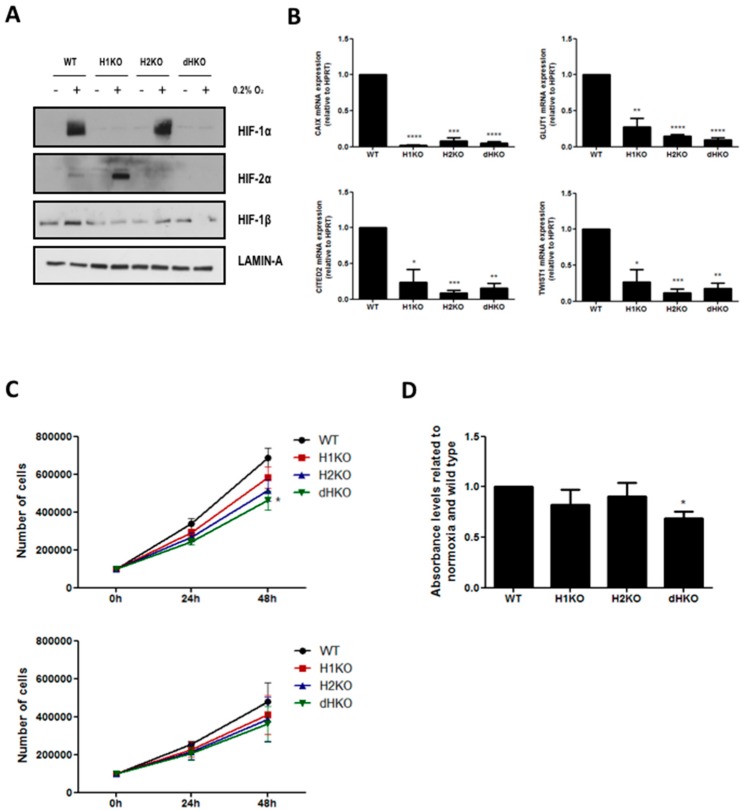
(**A**) Western blot of HIF-1α, HIF-2α and HIF-1β expression in H1299 cells under normoxic (21%) and hypoxic (0.2%) conditions. Lamin A was used as loading control. (**B**) mRNA expression of hypoxia-inducible transcription factors (HIF) target genes CAIX, GLUT1, CITED2 and TWIST1 after 24 h hypoxia. HPRT mRNA was used for normalization. (**C**) Automated cell counting of H1299 cells under normoxia (upper) and hypoxia (lower) at 24 h and 48 h after seeding. (**D**) Hypoxia tolerance was measured by crystal violet staining assay after 5 days of hypoxia incubation (0.2%). Asterisks indicate statistical significance (* *p* < 0.05; ** *p* < 0.01; *** *p* < 0.001; **** *p* < 0.0001).

**Figure 2 cells-08-00045-f002:**
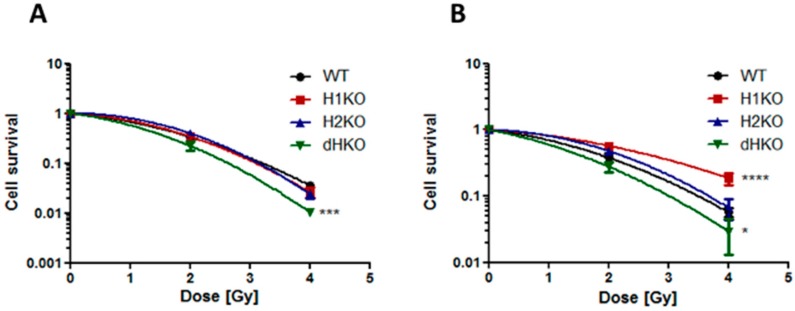
Clonogenic cell survival of H1299 cells irradiated under (**A**) normoxia (21% O_2_) and (**B**) hypoxia (0.2% O_2_). Surviving fraction was normalized to vehicle control. Average ± SD of three independent biological repeats is shown. Asterisks indicate statistical significance (* *p* < 0.05; *** *p* < 0.001; **** *p* < 0.0001).

**Figure 3 cells-08-00045-f003:**
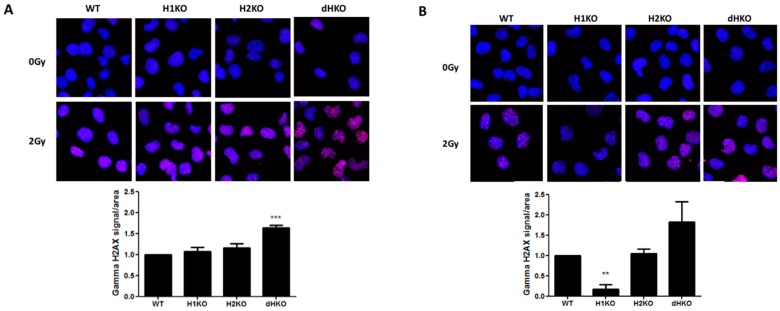
Representative merged fluorescent images of γ-H2AX foci (red) and nuclei (blue) upon irradiation under (**A**) normoxia and (**B**) hypoxia in H1299 cells. Quantification of γ-H2AX staining in cells. Average ± SD of three independent biological repeats is shown. Asterisks indicate statistical significance (** *p* < 0.01; *** *p* < 0.001).

**Figure 4 cells-08-00045-f004:**
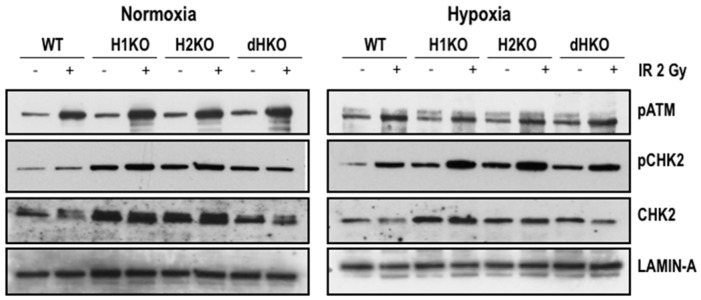
Western blot of pATM, pCHK2 and CHK2 expression in H1299 cells irradiated with 2Gy under normoxic (21%) and hypoxic (0.2%) conditions. Lamin A was used as loading control.

**Figure 5 cells-08-00045-f005:**
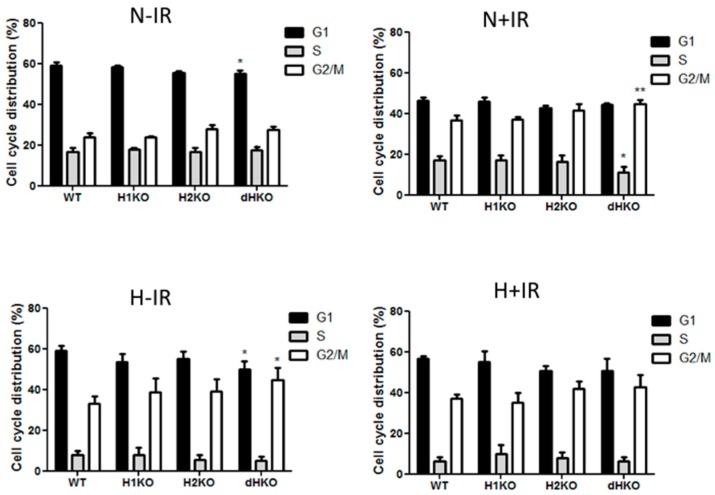
Flow cytometry analysis of cell cycle distribution of H1299 cells after 2Gy radiation under normoxic (21%) and hypoxic (0.2%) conditions. Fraction of cells in G1, S and G2-M phase in non-irradiated cells under normoxia and hypoxia and 4 h post-irradiation in normoxic and hypoxic cells. Average ± SD of three independent biological repeats is shown. Asterisks indicate statistical significance (* *p* < 0.05; ** *p* < 0.01).

**Figure 6 cells-08-00045-f006:**
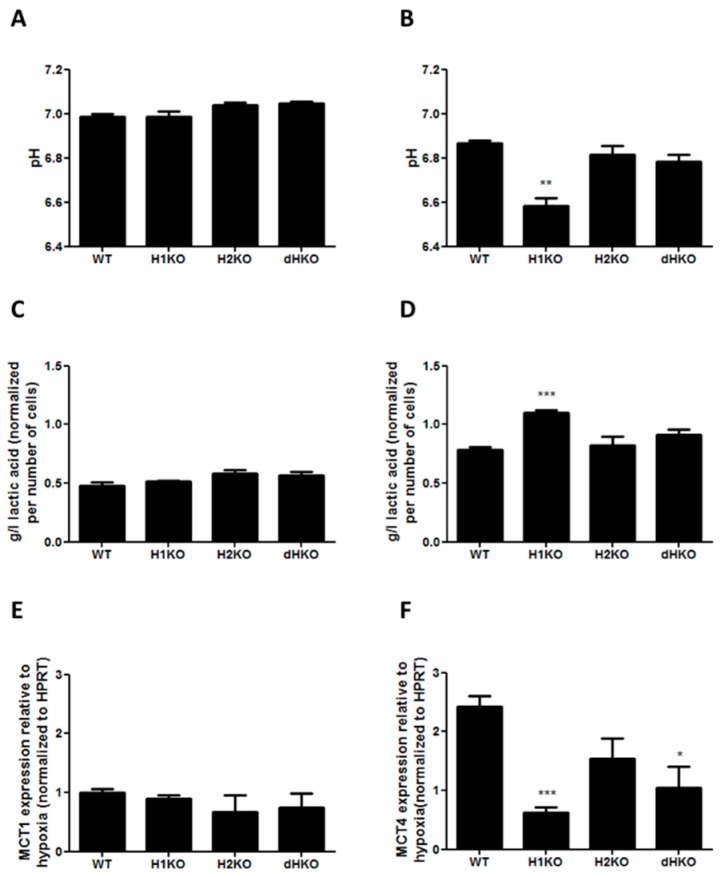
(**A**,**B**) Measurement of extracellular pH and (**C**,**D**) extracellular L-lactic acid accumulation upon normoxia (left panels) or hypoxia (right panels). Values were normalized to total cell number. mRNA expression of HIF target genes MCT1 (**E**) and MCT4 (**F**) after 24 h hypoxia incubation. HPRT was used as housekeeping gene to normalize expression data. Asterisks indicate statistical significance (* *p* < 0.05; ** *p* < 0.01; *** *p* < 0.001).

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
