# Peer review of "HIF-1α and HIF-2α Differently Regulate the Radiation Sensitivity of NSCLC Cells"

_cells, 2019, doi:10.3390/cells8010045_

Round 1

Reviewer 1 Report

The work by Roig et al utilizes Crispr-Cas9 invalidation of the two HIF-alpha subunits and evaluates the impact of the invalisation of each of them as well as of the combined invalidation of NSCLC cell sensibility. The work is well designed and the results are sound.

However, it is difficult to reconcilaite the fact that the double KO is the most sensitive with any of the results regarding the metabolism. Hence, HIF is protective through another mechanism. This is not clearly mentioned, nor discussed in the manuscript.

Minor comment : the Abstract should indicate more clearly that it is probably through the overexpression of the HIF-2alpha subunit that the invalidation of the 1alpha subunit rendered the cells more resistant.

Author Response

We would like to thank the reviewer for the kind words and the in-depth analysis of our manuscript.

Several experiments were performed assessing the effect of HIF1-a knock out (H1KO), HIF2-a knock out (H2KO) or double-HIF1/2a knock out (dHKO) on metabolism. More specifically: extracellular pH and lactic acid determination; assessment of MCT1 and MCT4 expression levels and finally OCR and ECAR cellular respiration experiments. The ph and lactic acid experiments only show relevant differences in H1KO cells under hypoxic conditions, leading to a protective phenotype upon radiation. Also, our experiments indicate relevant differences only for the H1KO cells (gain in glycolytic reserve) in support of the protective phenotype upon radiation, possibly mediated by the compensatory increase of HIF-2α. Referring to the minor comment, we will highlight more clearly in the abstract the radioprotective effect of HIF2-a overexpression in H1KO cells.

In contrast, the dHKO cells demonstrated increased radiation sensitivity, possibly explained by the overall reduction in glycolytic profile, although this did not lead to changes in pH or Lactic acid production. So, indeed HIF can be protective by other mechanisms, such as the ones mentioned in the discussion section: lines 309 and 310, including cell cycle arrest, apoptotic activity and VEGF signaling. Moreover, other investigations showed that HIF mediates the conversion of different metabolites into others. For instance, Stegen, et al. have demonstrated that HIF-1a mediates the conversion of glutamine to glutathione, preserving redox balance by inhibiting ROS (Stegen, et al. 2016). Active HIF would thereby reduce the levels of ROS due to the accumulation of glutathione in cells, leading to a radioprotective effect. Therefore, HIF-a proteins participate in many ways regulating cellular metabolism, which leads to variable levels of ROS or other relevant molecules involved in radiation response. 

We extended the discussion explaining these possible mechanisms.

Reviewer 2 Report

This is an interesting study comparing long term depletion of HIFs in response to IR and/or hypoxia. The results show that HIF-1alpha depletion results in radioresistance, possibly due to increases in HIF2alpha expression. Although ATM signalling seems to be fine, DNA damage marks such as H2Ax are lost in HIF-1alpha KO.

Although the data presented is of very good quality, these findings do raise a question regarding the approach of KOs. Do the authors find similar results with regards with H2Ax foci with short term depletion of HIFs? Also, was this observed in several clones and/or sgRNAs used? These are important considerations when using CRISPR mediated depletions.

Some discussion on where the defect is occuring in HIF-1alpha depleted cells would be ideal.

Some minor comments:

Is HIF-1beta also affected by HIF-1alpha depletion?

The compensatory effect by elevated HIF-2alpha although common could be cell type specific. Some discussion added would be good.

The effects on cell cycle are minor and should not be over-interpreted, especially after this approach since adaptation has occured.

Author Response

We would like to thank the reviewer for the kind words and the in-depth analysis of our manuscript. Indeed, we did not perform experiments upon short-term depletion of HIFs in this study. Previously, we have shown (Rouschop et al, PNAS 2013) using shRNA directed against HIF-1α, a reduced hypoxia tolerance and proliferation rates upon hypoxia, the latter similar as our results (Fig 1C). However, no differences in growth delay were observed upon irradiation between HIF-1α knock-down and wt cells, possibly explained by the residual HIF-1α expression. For this reason, we decided to generate a complete knock-out using the CRISPR/CAS9 system. We observed a more radioresistant phenotype for H1KO upon irradiation under hypoxia, while a sensitization for the dHKO cells using clonogenic assays as read-out (Fig 2). These results were indeed confirmed using gH2AX staining (Fig 3). Additionally, experiments in an independent clone confirmed the clonogenic assay results (Fig S4). A confirmation with gH2AX in the independent clones was not performed, since we believe the experiments with independent clones using clonogenic survival – our most important read-out – is already undisputable validation.

Discussion will be added where the depletion is occurring in HIF-1α knock-out cells.

HIF-1β expression levels are reduced for every knock-out cells (Fig 1A), not only by HIF-1α depletion. We confirmed that the gRNAs used for targeting HIF-1α and HIF-2α did not overlap with the HIF-1β locus. The HIF-1β reduction might be explained by the transfection procedure, since the similar effects found in all knock-out cells.

We will include an explanation for the implications on cell-type specificity for our phenotype on HIF2 upregulation in the discussion section.

We do agree that the effects on cell cycle are minor and softened the discussion and conclusion to prevent over-interpretation.

Round 2

Reviewer 2 Report

The authors have addressed the majority of my concerns given the limited time available. It would be good to include statistics on the new figure Fig. S4

Author Response

Unfortunately, experiments using the independent clones were performed only once in order to validate the KO cells, that is why statistics were not performed.